# Temporal aspects of unrealistic optimism and robustness of this bias: A longitudinal study in the context of the COVID-19 pandemic

Kamil Izydorczak[1]*, Karolina Antoniuk[1], Wojciech Kulesza[2], Paweł Muniak[2], Dariusz Dolinski[1]

**1** Faculty of Psychology in Wroclaw, SWPS University of Social Sciences and Humanities, Wroclaw, Poland, **2** Warsaw Faculty, Centre for Research on Social Relations, SWPS University of Social Sciences and Humanities, Warsaw, Poland

* kizydorczak@swps.edu.pl

**Data Availability Statement:** All data files are available from the Open Science Framework database (https://osf.io/4c3kr/?view_only=b8c01be2d17c4d8f892ba567b78d18f5).

## Abstract

Numerous studies on unrealistic optimism (UO) have shown that people claim they are less exposed to COVID-19 infection than others. Yet, it has not been assessed if this bias evolves; does it escalate or diminish when the information about the threat changes? The present paper fills this gap. For 12 months 120 participants estimated their own and their peers' risk of COVID-19 infection. Results show that UO regarding COVID-19 infection is an enduring phenomenon–It was the dominant tendency throughout almost the entire study and was never substituted by Unrealistic Pessimism. While the presence of UO-bias was constant, its magnitude changed. We tested possible predictors of these changes: the daily new cases/deaths, the changes in governmental restrictions and the mobility of participants' community. Out of these predictors, only changes in governmental restrictions proved to be significant- when the restrictions tightened, UO increased.

## Introduction

Numerous psychological studies have demonstrated that optimism is generally associated with better emotional, social, and task-related functioning. Optimism increases the chances of achieving success [1–3] and is associated with better physical health [4, 5].

Optimism can manifest itself in social comparisons [6]. Specifically, we may believe that fate is kinder to us than to others. In such a case, both positive and negative implications can be expected. This type of comparative optimism can help individuals to maintain a high level of well-being while also making them too carefree or reckless. This bias is referred to as unrealistic optimism (UO). In the current work, we present a study that is, to the best of our knowledge, the longest longitudinal study examining UO in the context of a real-world, enduring threat.

### Unrealistic optimism bias

Neil Weinstein was a forerunner in demonstrating that "people believe that negative events are less likely to happen to them than to others, and they believe that positive events are more

**Funding:** This research was supported by grants: 1. RID (Regionalna Inicjatywa Doskonałości Mazowsza - Regional Excellence Initiative for Masovian District - https://www.gov.pl/web/edukacja-i-nauka/regionalna-inicjatywa-doskonalosci): "Unrealistic optimism in the age of pandemic. Health research and ensuring safety for the inhabitants of Mazovia district" granted to Dariusz Dolinski (number: 2020/2). 2. The Polish National Agency for Academic Exchange (NAWA - https://nawa.gov.pl/) within the Urgency Grants programme granted to Wojciech Kulesza (number: PPN/GIN/2020/1/00063/U/00001). 3. SWPS University of Social Sciences and Humanities - internal grant from FRBN programme (Fundusz Rozwoju Badań Naukowych - Research Development Fund) granted to Wojciech Kulesza (number: 41/2022/FRBN/E) The funders had no role in study design, data collection and analysis, decision to publish, or preparation of the manuscript.

**Competing interests:** The authors have declared that no competing interests exist.

likely to happen to them than to others" [7]. Weinstein named this phenomenon "unrealistic optimism" (UO).

In their seminal article, Taylor and Brown [8] proposed treating UO as an instance of so-called positive illusions, the main purpose of which is to reduce stress and anxiety. From this perspective, one may conclude that UO helps people to cope with potentially threatening experiences.

Indeed, numerous studies have shown that unrealistic optimism appears in a range of contexts, some of which may pose a risk to individuals' wellbeing. For example, it has been found with respect to the probability of experiencing various diseases, such as alcoholism or heart attacks [9], breast cancer in women, and prostate cancer in men [10].

## Unrealistic pessimism

Although people are usually unrealistically optimistic, this is not always the case. For example, Dolinski and colleagues [11] conducted a study on Polish students one week after the tragic accident at the Chernobyl nuclear power station, when the radioactive cloud had arrived over the territory of Poland.

The participants were asked about the probability of falling ill with radiation sickness, and they judged that they were more vulnerable to the sickness compared to other individuals. The researchers termed this effect unrealistic pessimism (UP). A similar effect was noted by Burger and Palmer [12] after the 1989 California earthquake.

## Temporal aspects of unrealistic optimism/pessimism

Burger and Palmer [12] repeated their study three months later and discovered that the bias transitioned from UP to UO. Thus it seems that unrealistic pessimism is a short-term state.

Unrealistic pessimism can motivate an individual to take more preventive measures. In the Chernobyl study, unrealistic pessimists performed many more actions that could protect them from danger (for example, they tried not to leave the house and drank Lugol's liquid). However, since pessimism is associated with experiencing stress, anxiety, and a diminished sense of control, perhaps this state becomes too burdensome if it lasts too long. At some point, people may return to their initial belief that bad things will happen to other people rather than to them.

Another study, examining the temporal aspect of the UO bias, was conducted by Helweg-Larsen [13] in the context of the Northridge, California earthquake in 1994. In this longitudinal study, the author examined UCLA undergraduates one week after they experienced the earthquake and then in seven, consecutive waves during the following five months after the earthquake. Helweg-Larsen discovered that participants did not display an optimistic bias in respect to the earthquake and their realistic estimations persisted throughout the whole 5-month period. The author suggests that directly experiencing the disaster diminished individuals' sense of personal control over the particular event and therefore broke the illusion of invulnerability.

Of particular note, the participants displayed a UO bias in respect to other natural disasters (such as a flood) for the entire studied period. This fact might explain the difference in Burger and Palmers' results [12], because the authors of the earlier study did not ask questions about the earthquake specifically, but about 'natural disasters' in general.

## Unrealistic optimism in the COVID-19 era

The COVID-19 pandemic is a phenomenon that affects whole societies and similarly to the abovementioned Chernobyl disaster and earthquake, it was unexpected and almost completely

uncontrollable [11, 12]. Recent studies conducted in various parts of the world have observed UO regarding the possibility of contracting the coronavirus (for instance, in Iran, Kazakhstan, and Poland [14], in Romania and Italy [15] and in the USA and UK [16]).

It is noteworthy that the dynamic of the COVID-19 pandemic is different from the afore-mentioned catastrophes. In the case of a nuclear power plant explosion or an earthquake, the real threat rapidly peaks and then diminishes. In the case of the coronavirus pandemic, the situation endures for months, and now years, with fluctuations in the level of threat. These fluctuations are signaled both by objective data (cases, deaths, etc.) and political decisions (lockdowns, border closures, etc.). The question is how these changes in the situation may affect the level of UO.

## The goal of the study

Summing up, the present paper addressed three issues: (1) Is UO a robust phenomenon from a long-term perspective? (2) What is the relationship between changes in the level of danger and changes in bias? (3) What is the relationship between changes in the level of social isolation and changes in bias?

In detail, we were interested in whether UO disappears, turns into UP, or changes its magnitude. These changes may emerge when the media reports about the development of the pandemic and its severity, especially about increasing numbers of infected people and deaths from COVID-19.

People may also estimate the severity of the pandemic by observing the management of the pandemic by governmental bodies. Stricter restrictions (e.g., the introduction of lockdowns) may signal that "the situation is dangerous". Additionally, a liberalization of the rules of social coexistence (e.g., the opening of schools, shops, and restaurants, or by allowing fans to watch matches in stadiums) may signal that "it's safe".

Last but not least, Unrealistic Optimism may also be influenced by the cognitive availability of others' protective measures [17, 18]. If so, we might expect the magnitude of UO bias to change depending on how often we witness other people's behaviors. The bias could be stronger when we remain in household isolation, observing the behaviors of only a few close relatives and significant others.

## General method

### Participants

The study was conducted among Polish employees of an international corporation located in Wroclaw city (around 700,000 residents).

The sample consisted of 120 participants with university degrees (64 men and 56 women) aged 25–45 ($M_{age}$ = 33.64, $SD_{age}$ = 5.68) who agreed to answer a questionnaire. All participants worked in the same telecommunication company and on the same site for the whole period of the study. During most of the study, participants worked online. All participants held job positions related to computer programming.

The sample size was determined via feasibility criteria. The authors had to rely on the limited number of available participants, especially in the light of the rapidly evolving pandemic which forced the research team to organize the study as soon as possible. As a result, it was concluded that 120 participants in a one-condition, repeated-measures design was sufficient to detect meaningful effects. The results of the study were supplemented by a simulation-based power analysis [19]. The analysis indicated that assuming the obtained pattern of the means, the design of the study provided a power of 1-$\beta$ = 1 for $\alpha$ = .05. The simulation also indicated

that n = 33 would be sufficient to detect main effects and n = 44 would be sufficient to detect interaction effects with a power of 1-$\beta$ = .9 and $\alpha$ = .05.

All participants provided informed consent to participate in the study. Participation was fully voluntary, and participants did not obtain remuneration in any form. Since one of the authors was a contractor in the participants' workplace, high-quality data-gathering was ensured and all participants who entered the study participated in all of the waves (there were no dropouts). The study was reviewed and approved by the local [due to anonymity, further details to be provided] ethics committee (opinion number: 03/P/04/2020). Informed consent was obtained from all participants before enrollment in the procedure and data collection.

## Procedure

All of the data were collected via an online survey. The database, along with the code for reproducible analyses and figures is publicly available on Open Science Framework (https://osf.io/4c3kr/https://osf.io/4c3kr/?view_only=b8c01be2d17c4d8f892ba567b78d18f5).

The data collection started when the first cases of COVID-19 were confirmed across many European countries, but before the first case was confirmed in the participants' country of residence. This first research wave (out of 16) was conducted on 03/01/2020. The second wave was conducted one day after the first confirmed case of COVID-19 infection in Poland on 03/04/2020. The third wave was conducted on 03/06/2020, 4 days after the WHO announced COVID-19's pandemic status.

The dates of the waves were chosen to coincide with the "milestones" of the pandemic (rapid increase/decrease in contractions or deaths). Data collection stopped exactly 12 months after the first measurement and—more importantly—when the COVID-19 vaccination for the general population became available in Poland.

Finally, 16 waves of data collection were conducted on the following dates: 03/01/2020; 03/04/2020; 03/16/2020; 04/23/2020; 05/26/2020; 06/16/2020; 06/19/2020; 08/07/2020; 09/17/2020; 10/07/2020; 10/15/2020; 12/06/2020; 01/05/2021; 01/27/2021; 02/16/2021; and 03/03/2021.

R programming language was used to prepare, analyze, and visualize the data [20], along with the "tidyverse" package [21] and "afex" package [22].

## Risk and unrealistic optimism

In each wave, the participants were asked to answer two questions assessing the perceived risk of COVID-19 infection:

1. What is the probability that you will be infected with the novel coronavirus?

2. What is the probability that an average person of your age and gender will become infected with the novel coronavirus?

The respondents rated their answers on an 11-point scale (1 = Absolutely impossible; 11 = Absolutely certain).

These two questions served as a measure of the subjectively perceived risk of COVID-19 contraction for "Self" ($Risk_{Self}$) and "Others" ($Risk_{Others}$).

The measure of UO was obtained by subtracting the risk estimate for "Self" from the estimate for "Others". We called this measure "Comparative Index" ($C_{index} = Risk_{Others} - Risk_{Self}$). A positive score indicated that the person estimated their chances to be lower than others, therefore exhibiting UO. A negative score indicated that the person exhibited UP. A score of "0" would indicate a lack of both biases.

At the end of each round, participants were asked to provide their unique code consisting of the first letters of their parents' names and the number of their month of birth (e.g., TD07). This procedure enabled us to track the scores for the entire 12 months of the study.

Additionally, since vaccines against COVID-19 became available to the public, during the last two waves, the participants were asked about whether they were vaccinated (we observed that no participants were vaccinated) and whether they intended to get the shot when they become eligible for it. It turned out that 71.7% of the respondents were eager to get vaccinated as soon as vaccines became available for their demographic. Because of the lack of sufficient variance in the results, we decided not to analyze the relationships concerning vaccine-related variables.

## Primary analysis

In the primary analysis, we aimed to examine the changing patterns of UO and how they might be related to the changes in objective data regarding the COVID-19 pandemic (the number of daily cases and deaths). The number of new COVID-19 cases and deaths was obtained from the "Our World in Data" website [23]. We considered the numbers for the entirety of Poland. This information is the most reliable and consistent and they are derived directly from official governmental announcements.

Three research questions were posed:

1. Will we observe a main effect of Unrealistic Optimism; specifically, will $Risk_{Others}$ be higher than $Risk_{Self}$?

2. Will we observe a main effect of the waves? Will the $Risk_{Self}$ and $Risk_{Others}$ estimates change in accordance with the waves?

3. Will we observe a relationship between objective data (daily infections and daily deaths from COVID-19) and the estimations of risk and UO ($Risk_{Self}$, $Risk_{Others}$, and $C_{index}$).

## Results

Before addressing our research questions, we visually analyzed the distribution of the variables to detect possible outliers and to obtain an understanding of the data structure. After inspecting the box plots for $Risk_{Self}$ and $Risk_{Others}$ in all 16 waves, we concluded that (except for the first wave) there were no influential outliers. For that reason, we assumed that the differences between the means reflected the differences in the central tendencies and we performed no outlier deletion.

While visually inspecting the histograms for $Risk_{Self}$ and $Risk_{Others}$, we concluded that the distributions significantly differed from normal, forming either right-skewed or uniform shapes.

Upon inspecting the "Daily new cases" and "Daily deaths" variables, we discovered that the distribution was exponential in shape, which is a pattern that was expected in the case of the rising pandemic.

**$Risk_{Self}$ and $Risk_{Others}$ vs. waves.** To determine whether the risk estimates for "Self" and "Others" varied across waves, we conducted a 2 ("Self" *vs.* "Others") * 16 ("Waves") two-way, between-subject ANOVA with "Risk" as a dependent variable.

We found a strong main effect of UO ($F[1, 119] = 101.41$, $p < .001$, $\eta_p^2 = .46$). The average estimate of "risk" for "self" was significantly lower ($M_{risk\_self} = 4.77$, $SD_{risk\_self} = 3.33$) than for "others" ($M_{risk\_others} = 6.18$, $SD_{Risk\_others} = 3.15$) meaning that there was a main tendency for estimating own risk as lower than others' risk (UO).

Similarly, waves proved to differentiate the estimates of "Risk" ($F$[5.10, 606.32] = 52.06, $p <$ .001, $\eta_p^2$ = .3). The lowest estimates were observed in the first wave (01.03.20) ($M_{risk\_1}$ = 1.76, $SD_{risk\_1}$ = 1.14). The highest "risk" estimates were observed in the fifth wave (26.05.20) ($M_{risk\_5}$ = 7.10, $SD_{risk\_5}$ = 2.98), meaning that just before the first case of COVID-19 was reported (first wave), the risk estimates were lowest and were close to the "Absolutely impossible" point. The highest perceived threat was noted approximately three months after the first COVID-19 case in Poland.

The interaction between "Self/Others" estimates and "Waves" also proved to be significant ($F$[7.39, 879.18] = 10.92, $p <$ .001, $\eta_p^2$ = .08). To investigate this interaction, a contrast analysis was performed.

We tested estimates of risk for "Self" and "Others" pairwise in each of the 16 waves. UO (indicated by significantly higher risk estimates for "Others" than "Self") was found in all of the waves except for the first, fourth, and fifth. It should be noted that these waves were associated with either the lowest estimates of risk (Wave 1) or the highest estimates of risk (Wave 4 or Wave 5). Additional support for the nearly constant presence of UO is the proportion of responses indicating comparative optimism ($C_{index}$>0) and comparative pessimism ($C_{ndex}$<0). In all waves, par the aforementioned 1, 4 and 5, there were more comparative optimists than pessimists. In the last wave, this advantage was the biggest: 50.83% of responses indicated comparative optimism ($C_{index} > 0$) while only 6.67% indicated comparative pessimism ($C_{index} < 0$). The average composition of responses for all 16 waves was: comparative optimism ($C_{index} > 0$) = 36.46%, comparative pessimism ($C_{index} < 0$) = 13.07% and unbiased ($C_{index} = 0$) = 50.47%. See detailed results for all waves in the Supporting Information section and in the online repository (https://osf.io/4c3kr/).

In summary, the UO effect was present during the entire first year of the pandemic, except for brief periods when the estimates of risk were the most extreme (see the detailed table of contrast effects in the Supporting Information section or in the OSF repository: https://osf.io/4c3kr/https://osf.io/4c3kr/?view_only=b8c01be2d17c4d8f892ba567b78d18f5).

Additionally, we decided to investigate the presence of time trends in $Risk_{Self}$, and $Risk_{Others}$ using autocorrelation tests. This method is advisable when we want to detect whether there is a consistent (stable or seasonal) pattern in our longitudinal variable or if we are observing random changes [24]. We used the 'ACF' function from '*nlme*' package [25] (Pinheiro et al., 2022) in the R programming language [20]. See Fig 1 for a visualization of the autocorrelation patterns.

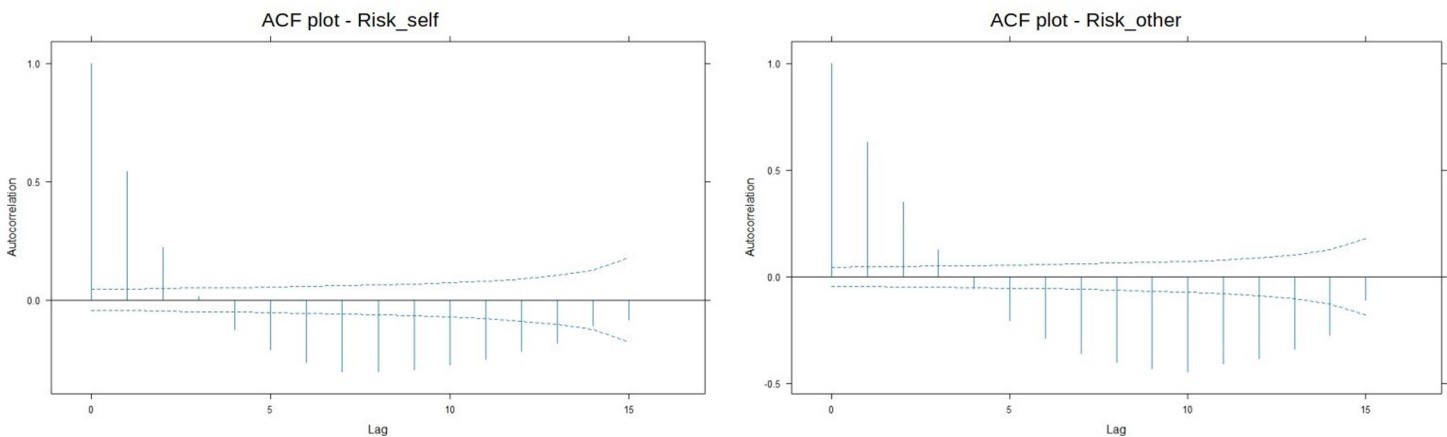

**Fig 1. ACF plot for Risk_Self and Risk_Others.** The blue line represents the cutoff points for correlations that are significant at p = .05.

In the graph we can observe that nearly every time point correlates with the previous ones, which indicates a strongly pronounced time-trend with little to no randomness in the pattern. The pattern signals a possible seasonality in the changes in risk estimates.

**Daily cases and daily deaths vs. Risk$_{Self}$, Risk$_{Others}$ and C$_{index}$.** To test whether the risk estimates and C$_{index}$ correlated with daily cases and daily deaths, we performed Kendall's tau tests. The results are presented in Table 1.

All the correlations were significant, and their directions indicated the positive relationship of risk estimates with daily cases and daily deaths; however, the relationships were weak. If not for the relatively large sample of observations, these correlations would not be substantially different from "0", and it is hard to acknowledge their practical or theoretical significance.

However, upon inspecting the visualized data regarding the relationship between risk estimates and daily cases/deaths, we observed a pattern of nonlinear relationships. The pattern became especially clear when cases and deaths were transformed to the logarithmic scale (and such a transformation is advisable for exponential distributions). See Figs 2 and 3 for details.

For both cases and deaths, the risk estimates initially increased. However, there was a "breaking point" at which the estimates for "risk" began to decrease with each higher order of magnitude of cases and deaths.

In the relationship between the C$_{index}$ and daily cases and deaths, we identified no such pattern—the C$_{index}$ fluctuated erratically as the number of cases and deaths increased (see the Supporting Information section or the OSF repository for visualizations: https://osf.io/4c3kr/?view_only=b8c01be2d17c4d8f892ba567b78d18f5https://osf.io/4c3kr/).

## Secondary analysis

While answering the first two research questions, we established that the magnitude of UO varied with the waves of the studies. Although UO was almost always present and was never substituted with UP, it was stronger during certain waves and weaker during other waves.

While addressing the third question, we established that changes in daily cases and deaths were not sufficient to explain the differences in the magnitude of UO.

According to the motivational explanations of Unrealistic Optimism, people exhibit it, because it helps them to cope with an ongoing or predicted threat [17, 18]. Our results suggest that there is almost no relation between the objective level of threat and UO, which casts doubt on the motivational roots of UO during the COVID-19 pandemic.

However, objective measures such as official statistics may not be the only or the most important source of information from which people may infer the level of threat. We concluded that another such source may be the strictness of COVID-19 preventative policies. First, this is because changes in these policies noticeably affected the lives of individuals and second, due to intensive information campaigns, they were salient,.

**Table 1. Correlation coefficients (r$_\tau$) between COVID-19 daily cases/deaths, risk estimates for "Self" and "Others", and intensity of UO.**

| n = 1920 | Daily cases | Daily deaths |
|---|---|---|
| Risk$_{Self}$ | $r_\tau = .04, p = .023^*$ | $r_\tau = .04, p < .014^*$ |
| Risk$_{others}$ | $r_\tau = .11, p < .001^{***}$ | $r_\tau = .08, p < .001^{***}$ |
| C$_{index}$ | $r_\tau = .10, p < .001^{***}$ | $r_\tau = .07, p < .001^{***}$ |

*—significant at $p < .05$,

***—significant at $p < .001$

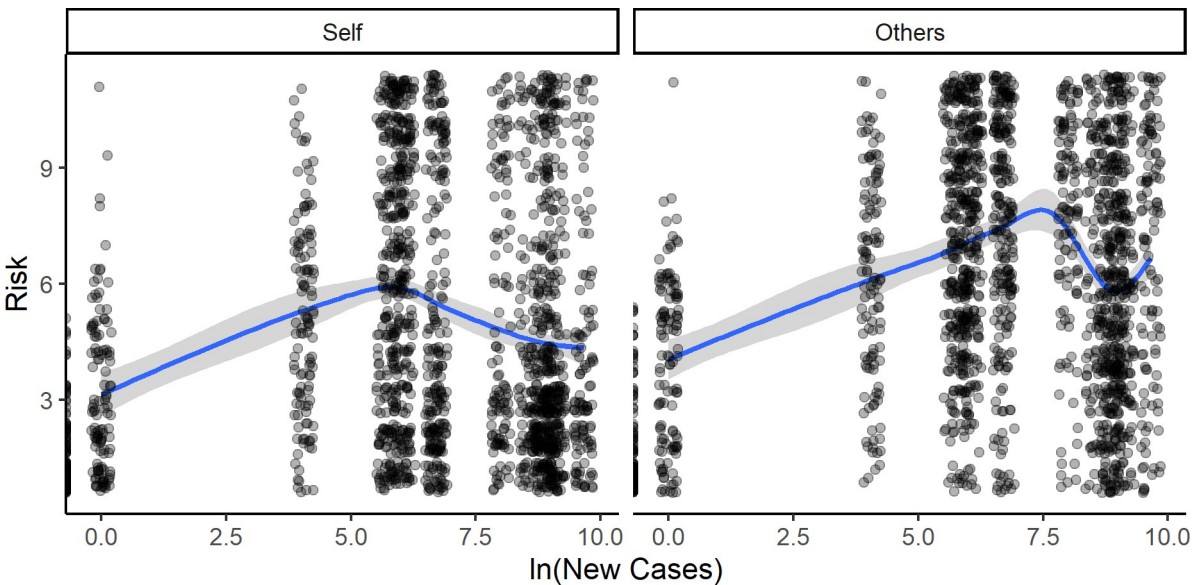

**Fig 2.** Natural logarithm of daily new cases vs. Risk$_{Others}$ (left panel) and Risk$_{Self}$ (right panel)—Visualization of locally weighted regression ('loess').

Assuming the motivational explanation of UO, we should expect UO levels to be higher when the restriction policies are stricter because they signal a stronger threat. To test this prediction, we computed a "Restrictions" variable, which captures the changes in governmental, anti-COVID policies.

Another possible time-varying factor that could influence the UO bias is the intensity of direct, social contacts. During the first year of the pandemic, people experienced different levels of social isolation–partly due to their own decisions and partly because of the changing

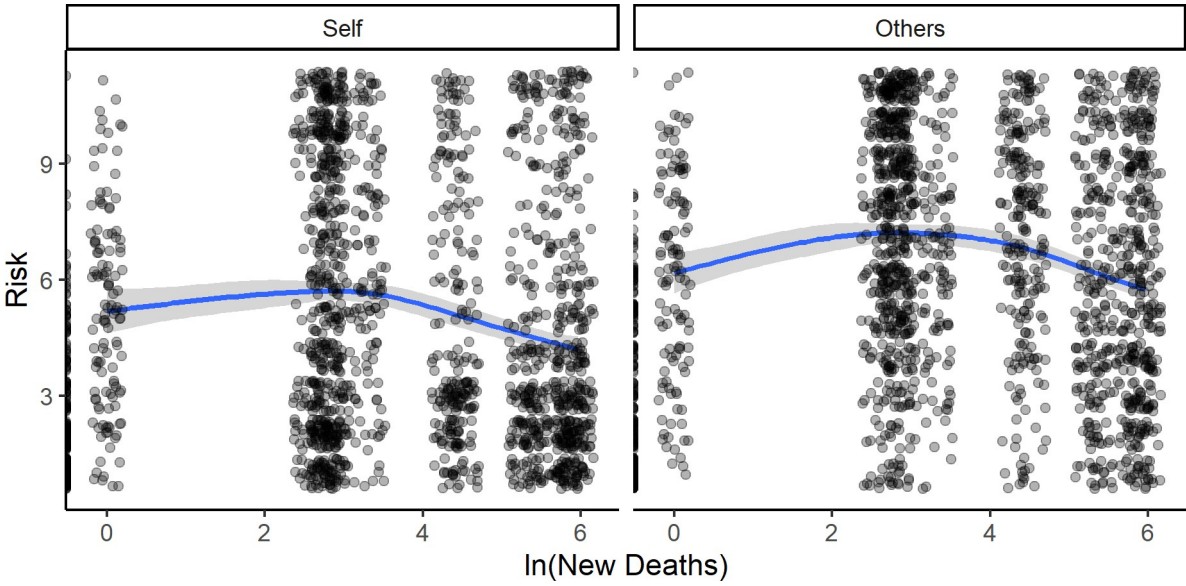

**Fig 3.** Natural logarithm of daily new deaths vs. Risk$_{Others}$ (left panel) and Risk$_{Self}$ (right panel)—Visualization of locally weighted regression ('loess').

laws and remote-work policies of their workplace. In line with the cognitive explanation for Unrealistic Optimism, people display it mostly due to asymmetry in cognitive perspectives–they are more aware of their own preventive measures than those of others and for that reason, they see themselves as less at risk [17, 18].

Assuming this explanation, communities should exhibit more Unrealistic Optimism when their members spend more time in their homes. With less direct contact with other people, the cognitive asymmetry should be further reinforced. To test this prediction we used data from Google Mobility Trends [26], which capture the changes in the time spent at home by communities in a given area.

## Restrictions

"Restrictions" was embedded in the timeline after data collection and it was a dichotomous variable reflecting the government policy at a given time in participants' residential area. "Restrictions" could exhibit two values: "easing" and "tightening".

We defined the "tightening" period as a time in which government officials announced new COVID-19 prevention restrictions. Usually, the announcements were made a few days ahead of enforcement (e.g., 10.03.2020 –ban on mass events, 31.03.2020 –introducing limits for customers in shops).

We defined the "easing" period as a time in which government officials announced and implemented laws that lifted some of the previous COVID-19-prevention restrictions. (e.g., 20.04.2020 –lifting the ban on recreational mobility and using public green spaces, 04.05.2020 –reopening of shopping malls).

The first wave of studies (01.03.2020) was left uncategorized, because at the time, there were no COVID-19 cases in Poland, no salient restrictions, and no clear message from the authorities.

This message suddenly changed upon the diagnosis of the first COVID-19 case (04.03.2020); thus, the second wave was classified as "tightening".

It is worth noting that in both the "easing" and "tightening" periods, the government decisions escalated: after the first "easing"/"tightening" announcement, typically another one occurred. Each period resulted in a reversal of the trend.

The details of the policies and rationale for coding the "easing" and "tightening" periods can be found in the Supporting Information section and the OSF repository (https://osf.io/4c3kr/)

We assumed that the "tightening" communication would send the general public a message that "the situation is serious" and "there is something to worry about". Assuming that UO is a means to cope with stressful events [27–29], we predicted that when restrictions were tightened, UO should be higher.

Analogously, we supposed that in the "easing" period, officials sent a comforting message: "things are getting better" and "you don't have to worry as much as you did". During one of the "easing" periods, the Prime Minister of Poland stated explicitly: "I am glad that we are less and less afraid of this virus. It is a good approach because it [COVID-19] is on the retreat" (01.07.2020).

## Social isolation in the community: Google mobility trends

To quantify the degree of social isolation in the community in which participants lived, we acquired the Google Mobility Trend score for Wroclaw County, Poland (the residential area of our respondents) during the days in which we conducted our waves of measurement.

We used the "Residential" score category, which calculates the change in the time spent at home among a given population [26]. The "Residential" score is calculated as the percentage change in time spent at home, using the first week of February 2020 as the baseline. The score for each day of the week is calculated based on the baseline value for that day of the week from 1–8 February 2020. In the first week of February 2020, there were no cases of COVID-19 in Poland and no regulations affecting citizens' everyday activity and mobility.

The Google mobility "Residential" score is a measure of overall time spent outside the household by members of the community from a given area. It is based on data provided by smartphones using Google software. Given that 78% of citizens in Poland are smartphone users and almost 90% of these users use the Android system, the score can be a good representation of actual mobility [30]. This might be especially true in highly urbanized areas, such as Wrocław, which has many students and white-collar workers as residents.

It is important to note that the Google Mobility Trends score does not directly relate to the behavior of our participants, but rather to their environment as a whole. On days when the "Residential" score was lower, the time spent outside the household within the whole community was longer. It means that participants were more likely to visit public places, and they were more likely to observe more people in these spaces. This feature makes the Google Mobility Trends score a particularly appropriate variable for measuring the overall intensity of face-to-face, social interactions. For the purpose of our hypothesis, we were looking for a measure that corresponds with the chances of directly observing other people's behaviors—we believe that the Google Mobility "Residential" score serves this goal well because it corresponds with the number of people "on the streets" at a given time.

## Restrictions vs. $Risk_{Self}$, $Risk_{Others}$ and $C_{index}$

To test whether changes in "Restrictions" could explain changes in absolute and relative estimations of COVID-19 infection risk, we conducted linear mixed-model analyses with "Restrictions" ("Easing" *vs.* "Tightening") as a fixed effect variable, individuals' ID as a random grouping factor and $Risk_{Self}$, $Risk_{Others}$, and $C_{index}$ as dependent variables. In each analysis, we used Type-III sum of squares and model terms were tested with likelihood-ratio tests. In each analysis, we allowed the slopes to vary by the "ID". Models were fitted with the ML method.

In addition to testing the fit, we also compared the "Restrictions" fixed effect models to models with "Waves" fixed effects, to establish whether "Restrictions" provide a better fit for the model than the "Waves" themselves. AIC and BIC were used to compare the models. JASP ver. 0.16.0 [31] was used in the calculations and visualization. The reproducible analysis can be found in the OSF repository (https://osf.io/4c3kr/)

"Restrictions" proved to be a significant predictor for $Risk_{Self}$, ($\chi^2[2] = 42.27$, $p < .001$). The estimate marginal means for $Risk_{Self}$ were higher for the "easing" ($M_{easing} = 5.60$, $SE_{easing} = 0.19$) than for the "tightening" condition ($M_{tight.} = 4.56$, $SE_{ight,} = 0.19$). The fit statistics for the model were: $AIC = 9013.69$ and $BIC = 9046.66$ and were higher than those for the analogical model with "waves" fixed effects ($AIC = 9470.05$, $BIC = 9507.13$).

To summarize, "Restrictions" proved to predict $Risk_{Self}$ with a better fit than "waves"; when restrictions were tighter, the risk estimates for "Self" decreased. See Table 2 for detailed results of the analysis.

"Restrictions" proved to be a significant predictor for $Risk_{Others}$ as well, although the effect was weaker ($\chi^2[2] = 8.77$, $p < .001$). The estimate marginal means for $Risk_{Others}$ were higher for the "easing" ($M_{easing} = 6.67$, $SE_{easing} = 0.19$) than for the "tightening" period ($M_{tight.} = 6.33$, $SE_{ight,} = 0.18$. The fit statistics for the model were: $AIC = 8556.04$ and $BIC = 8589.01$ and were higher than those for the "waves" fixed effect model ($AIC = 8755.67$, $BIC = 8855.75$).

**Table 2. Summary of linear mixed-models with Risk$_{Self}$, Risk$_{Others}$ and C$_{index}$ as dependant variables, "Restrictions" as a fixed effect and "ID" as a random effect.**

| Dependent variable | Estimates for "Restrictions" | Mean "Easing" | Mean "Tightening" |
|---|---|---|---|
| Risk$_{Self}$ | Intercept = 5.08, SE = 0.17 | **M = 5.60**, | M = 4.56, |
| | $b$ = 0.52***, SE = 0.07, df = 119.99 | 95% CI [5.22, 5.97] | 95% CI [4.19, 4.93] |
| Risk$_{others}$ | Intercept = 6.50, SE = 0.18 | **M = 6.67**, | M = 6.33, |
| | $b$ = 0.17***, SE = 0.06, df = 1679.45 | 95% CI [6.3, 7.04] | 95% CI [5.97, 6.68] |
| C$_{index}$ | Intercept = 1.42, SE = 0.14 | M = 1.07, | **M = 1.76**, |
| | $b$ = -0.35***, SE = 0.07, df = 120 | 95% CI [0.78, 1,37] | 95% CI [1.43, 2.10] |

***—significant at $p < .001$, **bold** indicates higher mean value row wise.

Analogous to Risk$_{Self}$, "Restrictions" proved to predict Risk$_{Others}$ with a better fit than "waves" and the risk estimates for "Others" decreased when restrictions were tightened. See Table 2 for detailed results of the analysis and see Fig 4 for the visualizations of the changes in risk estimates over time as well as in different restriction periods.

Finally, we tested whether "Restrictions" can predict the C$_{index,}$ which is our measurement of the magnitude of UO bias. "Restrictions" proved to be a significant predictor ($\chi^2[2] = 20.29$, $p < .001$). The estimated marginal means for C$_{index}$ were lower for the "easing" ($M_{easing}$ = 1.07, $SE_{easing}$ = 0.15) than for the "tightening" period ($M_{tight.}$ = 1.76, $SE_{ight,}$ = 0.17). The fit statistics for the model were: $AIC$ = 9015.28 and $BIC$ = 9048.26 and were higher than those for the "waves" fixed effect model ($AIC$ = 9448.89, $BIC$ = 9548.97).

UO bias proved to be stronger in the "tightening" conditions and once again "restrictions" proved to be a valuable explanatory variable, providing a predictive model with a better fit than the plain repeated measures variable ("Waves"). See Fig 5 for visualization of the UO bias changes along with the waves and different "Restrictions" periods.

## Social isolation vs. Risk$_{Self}$, Risk$_{Others}$ and C$_{index}$

To test whether changes in "Residence Mobility" could explain changes in absolute and relative estimations of COVID-19 infection risk, we used an analogical, linear mixed-model strategy.

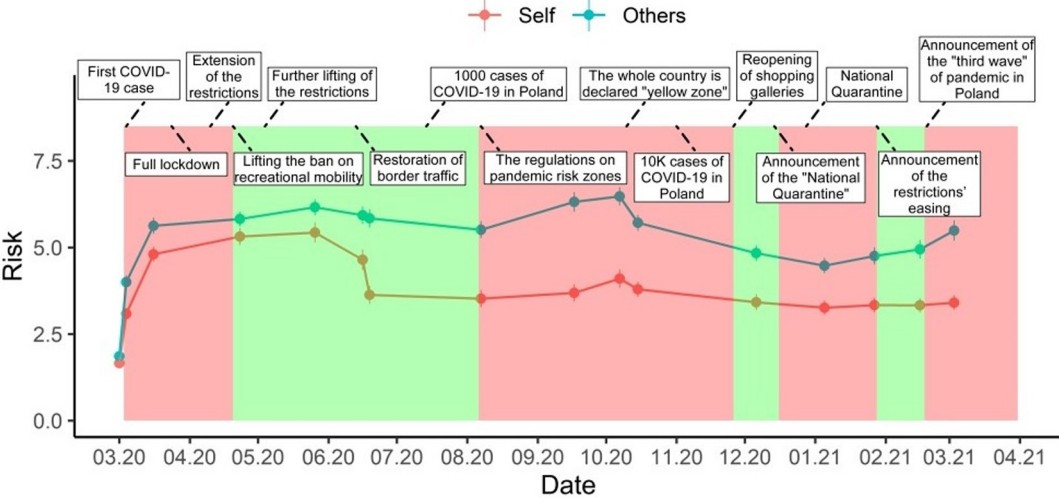

**Fig 4. Line plot of changes in risk estimates over time.** Each dot represents mean risk estimates for "Self" (blue) and "Other" (red) at a given time. Bars represent standard errors of means. Frames above the graph describe the most important events in the timeline of the pandemic.

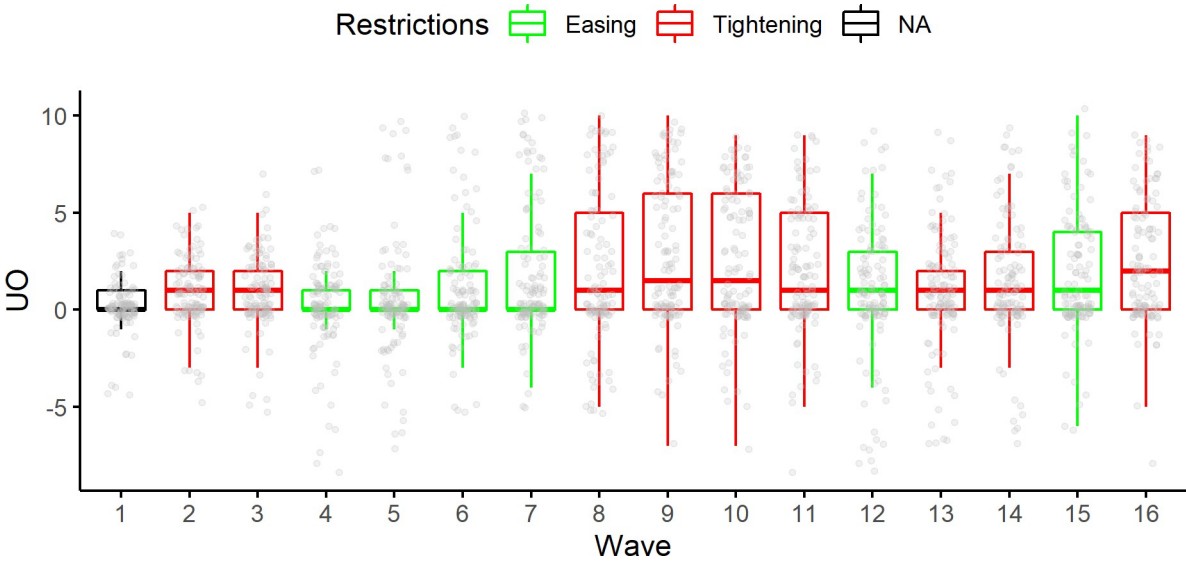

**Fig 5. Box-plot of $C_{index}$ distribution in all waves.** Jittered points represent the density of $C_{index}$ values. The color of the boxes represents the "Restriction" period in which the wave took place.

We included the "Residence" mobility score as a fixed effect variable, individuals' ID as a random grouping factor and $Risk_{Self}$, $Risk_{Others}$, and $C_{index}$ as dependent variables. In each analysis, we used Type-III sum of squares and model terms were tested with Satterthwaite approximation. In each analysis, we allowed slopes to vary by the "ID" variable. Models were fitted using REML.

"Residence Mobility" proved to be a significant predictor for $Risk_{Self}$ ($F$ [1, 1752.43] = 106.98, $p < .001$). The model indicates that higher "Residential Mobility" scores predict higher $Risk_{Self}$ estimations ($b = 0.10$, SE $< 0.01$, $t = 10.34$, $p < .001$), which means that the more time the community spent at home, the higher the $Risk_{Self}$ estimates of our participants.

"Residence Mobility" proved to be a significant predictor for $Risk_{Others}$ as well, ($F$ [1, 906.76] = 86.30, $p < .001$). The model indicates that higher "Residential Mobility" scores predict higher $Risk_{Others}$ estimations ($b = 0.08$, SE $< 0.01$, $t = 9.26$, $p < .001$), which means that the more time the community spent at home, the higher the $Risk_{Others}$ estimates of our participants.

Finally, we tested whether "Residence Mobility" predicts the magnitude of the UO bias. Assuming the cognitive explanation for UO, higher "Residential Mobility" scores should predict a higher $C_{index}$. However, it turned out that this relation is statistically non-significant ($F$ [1, 453.74] = 2.47, $p = .12$).

To summarize, when the amount of time spent at home rises within a community, the risk estimations made by the members of this community increase. This is true for both "Self" and "Others" estimations. However, "Residential Mobility" does not appear to be related to the magnitude of the UO bias.

## Discussion

We inspected two sources of information about the severity of the COVID-19 pandemic. The first source was objective data—the number of daily cases and deaths. The second source was governmental restrictions. While the numbers are abstract and difficult to interpret, restrictions are experienced directly and had a salient impact on the participants' lives.

We noted that the strength of the optimistic bias was almost independent of both the number of cases and deaths, however, the strength of unrealistic optimism did vary in accordance with the changes in policies by the state authorities.

When restrictions tightened, UO increased. The increase in UO took place mostly due the decrease in risk estimations for "Self". The estimates for "Others" also decreased during the "tightening" periods, but to a lesser degree.

Furthermore, we observed that although the degree of social isolation predicted both the risk estimates for "Self" and "Others" (the more contact, the less perceived risk), we did not find evidence for a relationship between social isolation and UO.

In light of the motivational explanation, UO is displayed during the COVID-19 pandemic because it is an extremely stressful situation and the more threat people experience, the stronger their UO bias.

The cognitive explanation for the UO suggests that people display UO because they are more aware of and are more concentrated on their own efforts to prevent COVID-19. The efforts of others are much less accessible.

Our study provides mixed support for both explanations. The purely motivational explanation is undermined by the lack of evidence for the relationship between UO and daily cases/deaths, which are a clear indication of the threat level. On the other hand, the purely cognitive explanation is also less plausible, considering the lack of evidence for a relationship between UO and social isolation.

The one factor that predicted the magnitude of UO was governmental restrictions and this factor can be interpreted in the light of both cognitive and motivational explanations. In fact, it can contain the elements of both mechanisms. On the one hand, the governmental restrictions can form stronger cues for threat than the objective numbers, affecting motivational mechanisms. On the other hand, in times of stronger restrictions, people were forced to take many additional, preventive measures, which required conscious effort and attention–this could reinforce the cognitive basis of the UO.

Our data should best be interpreted in conjunction with other studies conducted in the context of the COVID-19 pandemic. Correlational studies from multiple countries found a positive relationship between the gravity of the pandemic situation and the magnitude of UO [32], which supports a motivational explanation. On the other hand, a recent study by Vieites and colleagues [33] demonstrated experimentally that the cognitive availability of one's protective behaviors enhanced UO in the context of the COVID-19 pandemic, which provides support for a cognitive explanation.

It is worth noting that, as a general conclusion of our studies, the UO remained a dominant tendency throughout the whole first year of the pandemic. Contrary to the Burger and Palmer study [12], the bias was not present before the event was directly experienced (first wave) and contrary to Helweg-Larsen's study [13], it persisted even when the pandemic started to affect the studied population.

In comparison to natural disasters such as earthquakes, the COVID-19 pandemic is more pervasive and less directly experienced (while every human in the area feels the physical sensation of the earthquake, not everyone becomes infected with the virus and the virus itself is not visible with the naked eye). For that reason, the patterns of UO during the COVID-19 pandemic might differ significantly from patterns discovered in other contexts.

## Limitations and directions for future research

While the longitudinal design provides unique insights, it also comes with limitations. First, in longitudinal studies, it is impossible to account for every event in the lives of individuals,

which may have a significant influence on the results. We can assume that general patterns of results may remain unaffected (which is supported by the lack of outliers), but some experiences, such as illness or the death of relatives/acquaintances, might be shared by many participants at the same time.

Second, our explanation of the differences in the level of UO was focused on three time-varying factors and we cannot exclude that other longitudinal processes could influence UO. One such example is political events during the first year of the pandemic–in that particular year, the citizens of Poland took part in national parliamentary elections that were accompanied by various controversial decisions and organizational difficulties.

The third important limitation is the composition of our sample. It mainly consisted of young adults with higher education who worked in the same company. While such a sample is still more diverse in terms of age and gender than standard student-only compositions, it is worth noting that the scope of the generalization of our study could be limited. Moreover, the homogeneity of our sample might have a significant impact on the baseline level of their unrealistic optimism. It has been shown that lower age and higher education are associated with higher unrealistic optimism [34]. Judging by the aforementioned research, we could expect that our sample might have higher levels of unrealistic optimism and lower levels of unrealistic pessimism than the general population. Future studies might replicate our research while—at the same time—employing more demographic/medical data about the participants to assess possible important factors that might influence the pattern of results.

The last limitation concerns the score of Google Residential Mobility Trends. We acknowledge that this measure is an indirect indicator of the number of observed individuals and might not necessarily reflect the local phenomena, such as the number of interpersonal contacts in particular neighborhoods or stores. Moreover, it does not account for the observation of others' behavior via traditional media and social media.

The two proposed mechanisms (the increase in threat, followed by the intensification of coping, and the increase in cognitive accessibility asymmetries) are not mutually exclusive. They may also co-occur or even reinforce one another. Moreover, during a time of increased threat, the ego-serving potential of asymmetric cognitive accessibility may help in coping with a stressful environment. Future research should aim to clarify this issue—and more importantly—replicate our results under different threats (not COVID-19 related).

It would also be interesting to verify the dynamics of unrealistic optimism in other contexts; for example, in regards to a serious illness that has phases of improvement and deterioration regarding the patient's condition or during prolonged attempts by a woman to become pregnant. It would also be particularly important to investigate how the rise and fall of unrealistic optimism are related to people's different decisions.

Another fruitful direction for future research could be investigating possible interactions between ecological and internal factors in longitudinal settings. For example, it has been proved that the magnitude of unrealistic pessimism with respect to the risk of breast cancer is higher for women with more comorbidities [34].

If we could track risk-related changes in the environment along with changes in infra-personal factors (such as health status), we may be able to understand the mechanism and limitations of the relationships between UO and external circumstances.

## Supporting information

**S1 Table. Frequency of comparative optimism/pessimism/unbiased across waves.**
(DOCX)

**S2 Table. Contrast effects for ANOVA—RiskSelf and RiskOthers vs. waves.**
(DOCX)

**S1 Fig. Visualization of locally weighted regression ('loess'): Daily new cases vs. C_index.**
(TIF)

**S2 Fig. Visualization of locally weighted regression ('loess'): Daily new deaths vs. C_index.**
(TIF)

**S3 Fig. Timeline of the study.**
(PNG)

**S1 File. Calendar of restrictions.**
(XLSX)

## Author Contributions

**Conceptualization:** Kamil Izydorczak, Karolina Antoniuk, Dariusz Dolinski.

**Formal analysis:** Kamil Izydorczak, Karolina Antoniuk.

**Funding acquisition:** Wojciech Kulesza, Dariusz Dolinski.

**Investigation:** Kamil Izydorczak, Karolina Antoniuk.

**Methodology:** Dariusz Dolinski.

**Project administration:** Karolina Antoniuk.

**Supervision:** Wojciech Kulesza, Dariusz Dolinski.

**Validation:** Paweł Muniak.

**Visualization:** Kamil Izydorczak.

**Writing – original draft:** Kamil Izydorczak.

**Writing – review & editing:** Wojciech Kulesza, Paweł Muniak, Dariusz Dolinski.

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
