## [Decision Letter · Decision Letter 0]

17 Aug 2022

PONE-D-22-05224Temporal aspects of unrealistic optimism and robustness of this bias: longitudinal study in the context of the COVID-19 pandemicPLOS ONE

Dear Dr. Izydorczak,

Thank you for submitting your manuscript to PLOS ONE. After careful consideration, we feel that it has merit but does not fully meet PLOS ONE’s publication criteria as it currently stands. Therefore, we invite you to submit a revised version of the manuscript that addresses the points raised during the review process.

 Your article has been reviewed by one reviewer and their comments are appended below.  The reviewer has raised a number of concerns that require your attention. They state that the results need to contain more information about the participant characteristics in addition to age and sex only, and these differences in characteristics might need to be discussed further. In addition, the reviewer states that the overall reporting of the article could be improved.  Please note that we have only been able to secure a single reviewer to assess your manuscript. We are issuing a decision on your manuscript at this point to prevent further delays in the evaluation of your manuscript. Please be aware that the editor who handles your revised manuscript might find it necessary to invite additional reviewers to assess this work once the revised manuscript is submitted. However, we will aim to proceed on the basis of this single review if possible. 

We look forward to receiving your revised manuscript.

Kind regards,

Maria Elisabeth Johanna Zalm, Ph.D

Editorial Office

PLOS ONE

https://journals.plos.org/plosone/s/file?id=ba62/PLOSOne_formatting_sample_title_authors_affiliations.pdf".

Reviewers' comments:

Reviewer's Responses to Questions

**Comments to the Author**

1. Is the manuscript technically sound, and do the data support the conclusions?

Reviewer #1: No

2. Has the statistical analysis been performed appropriately and rigorously? 

Reviewer #1: Yes

3. Have the authors made all data underlying the findings in their manuscript fully available?

Reviewer #1: Yes

4. Is the manuscript presented in an intelligible fashion and written in standard English?

Reviewer #1: Yes

5. Review Comments to the Author

Reviewer #1: I thank the authors for their interesting contribution toward trying to better understand the pandemic. I agree that the pandemic is an excellent circumstance to study the phenomenon of unrealistic optimism (UO). The authors produced a longitudinal assessment of UO in a Polish cohort (all same employer), finding that UO increased in association with government restrictions (used as a proxy of potential risk perception). I very much appreciate the effort it took to complete this project. I hope that these comments (in no particular order) are useful.

1. The need for editing by a native-English speaker is completely understandable and recommended.

2. I recommend specifying the ethics committee that approved the project.

3. I think it would be easier on the reader if the dates of the wave were illustrated in a figure that also included the pandemic milestones as discussed on page 13.

4. A lot of statistical analyses could lead to autocorrelation, which could be addressed.

5. Did UP correlate with reporting likelihood of saying they would get vaccinated? Those with UP may be quite anxious to get vaccinated.

6. For the primary analyses, were local case counts and deaths used, or were those numbers for the country or world? How do you know all participants had the same information on COVID case numbers?

7. My concern (enough to prevent recommendation for publication until clarification) is that there appear to be no more data on participant characteristics besides age and sex. What were their health status, pre-existing conditions, political affiliation, occupation, and other factors that likely explain UO? Many people developed both RP and UP because of their health risk factors, and these effects are likely much, much greater than anything gleaned from government restrictions. It may also be that those with UO thought they were at lower risk during the government restrictions simply because they were not allowed to interact with others as much. And I remain unconvinced that Google mobility trends are a proxy for amount of social contact.

Overall this is a great idea and I appreciate the effort; the execution does not substantiate the conclusions at this time without more information on participant health status.

6. PLOS authors have the option to publish the peer review history of their article (what does this mean?). If published, this will include your full peer review and any attached files.

Reviewer #1: **Yes: **Michael Muehlenbein

---

## [Author Response · Author response to Decision Letter 0]

30 Sep 2022

Response letter

Dear Editor,

Below, we present the responses to your’s and reviewer’s comments. Please note that in the ‘track changes’ manuscript, all the new or thoroughly reviewed passages are highlighted with the green font color. Minor changes and language correction are included as ‘suggestions’ in ‘track changes’ mode. 

Comments to the Author

1. The need for editing by a native-English speaker is completely understandable and recommended.

Done! We have sent our revised version of the manuscript to another (different) proofreader who is a native speaker. We hope that the overall linguistic quality of the manuscript is improved. 

2. I recommend specifying the ethics committee that approved the project.

To preserve anonymity, in the first version of the manuscript, we decided not to include detailed information about the ethics committee. 

However, thanks to your comment we have realized that we could - while addressing your great suggestion - add a more precise ethical statement, which now reads (please see p.7):

“The study was reviewed and approved by the local [to be completed after acceptance] ethics committee (opinion number: 03/P/04/2020). Informed consent was obtained from all participants before enrollment in the procedure and data collection.”

The full ethical approval information has already been disclosed to the PLOS One editorial board and will be included in the final version (if accepted).

We are hoping that the provided changes and explanations are satisfactory. If not we would be more than happy to provide further changes according to your next suggestions.

3. I think it would be easier on the reader if the dates of the wave were illustrated in a figure that also included the pandemic milestones as discussed on page 13.

Thanks to your comment we realized that the readability of Figure 3 might be improved. As suggested, we added information about pandemic milestones in Figure 3. Moreover, we also created an additional visualization which includes dates and information about the milestones. It can be found in the Supplementary Materials (S6 Figure. Timeline of the study). 

4. A lot of statistical analyses could lead to autocorrelation, which could be addressed.

Thank you for bringing the topic of autocorrelation to our attention. You are absolutely right that these analyses enrich the picture of the results section, we have included them in the present version of the manuscript.

Since we were not sure which variables you would like to see analyzed we decided to examine the autocorrelation patterns for two main variables - risk estimates for "self, and "others". In the paragraphs on the analysis of time trends in the variables "Risk_self" and "Risk_others", we have added ACF graphs with their interpretation. 

Please see the changes on p.11. We hope that we have fully addressed your comment. However, if you find our work incomplete we would be more than happy to provide more analysis. In this case, we kindly ask for more details of the variables that you would like to see in the statistical model.

In line with our expectations, we found that all of our variables had high autocorrelation across almost the entire timeline. This indicates that instead of randomness or "white noise," we can speak of clear temporal trends in our data. Please note that in the case of our design, “autocorrelation” is a signal of the presence of a time-related pattern and it bears no negative meaning for the robustness of our analyses.

Thanks to your comment, we were also able to spot another crucial mistake in our description of the analysis. In the previous version of the manuscript, on p.10, we wrote, that we conducted a two-way between-within ANOVA, while in fact, our analysis is a two-way within-subject ANOVA. Please note that while auto-correlation can negatively influence the validity of between-subject comparisons (as described by Raadt, J. S. 2019, Journal of Methods and Measurement in the Social Sciences), it does not invalidate our analyses, which rely only on within-subject comparisons. 

5. Did UP correlate with reporting likelihood of saying they would get vaccinated? Those with UP may be quite anxious to get vaccinated.

Thank you for this important question and we fully agree that it should be addressed in the paper. 

It turned out that almost all of our participants were enthusiastic about vaccination, so we observed little variance. For this reason, we were unable to draw firm conclusions about the relationship between vaccine intention and other variables. In that regard, our sample is a good representation of the general trends in the population - In Poland, it was empirically established that the highest vaccine intention was found among people with university degrees, among the residents of the largest city and in the western part of the country. All of these characteristics apply to our participants.

In the current version of the manuscript, we explained why we did not conduct any analyses using the vaccine intention variable (see p.9) 

6. For the primary analyses, were local case counts and deaths used, or were those numbers for the country or world? How do you know all participants had the same information on COVID case numbers?

We used the number of deaths and cases on a national level. Thanks to your comment, we realized that the previous manuscript was missing this important information. We have added this clarification on p. 9. 

We have chosen not to regionalize these measures because regional data (whether for a voivodship or a city) were less propagated among the public.

Regarding the second question ("How do you know all participants had the same information on COVID case numbers?"), we can firmly assume that as long as the participants were interested in collecting information on the pandemic, they would have encountered the same numbers at each stage. In Poland, all mainstream and regional media gathered their information directly from official, central government announcements or indirectly from the Polish Press Agency. Information about the pandemic was neither highly localized nor divergent. Of course, we cannot rule out the possibility that some of the participants followed completely niche coverage (e.g., from sources denying the pandemic or engaging in conspiracy theories). However, this phenomenon would be marginal, and be unlikely to affect a significant portion of participants. Moreover, such cases would not invalidate conclusions about the intrapersonal changes we observe over time - this could be a much more serious problem in a non-randomized, between-subjects design.

7. My concern (enough to prevent recommendation for publication until clarification) is that there appear to be no more data on participant characteristics besides age and sex. What were their health status, pre-existing conditions, political affiliation, occupation, and other factors that likely explain UO? Many people developed both RP and UP because of their health risk factors, and these effects are likely much, much greater than anything gleaned from government restrictions. It may also be that those with UO thought they were at lower risk during the government restrictions simply because they were not allowed to interact with others as much. And I remain unconvinced that Google mobility trends are a proxy for amount of social contact.

Overall this is a great idea and I appreciate the effort; the execution does not substantiate the conclusions at this time without more information on participant health status.

Thank you for these insightful comments. Let us address each of your concerns separately.

“there appear to be no more data on participant characteristics besides age and sex.”

In addition to age and gender, we also mentioned that all survey participants had a university degree and all worked for the same telecommunications company in the same city (presumably, they all lived in the city or at least in the immediate vicinity).

In response to your comment, we added additional demographic information and added more detailed information (see p.6).

“What were their health status, pre-existing conditions, political affiliation, occupation, and other factors that likely explain UO? Many people developed both RP and UP because of their health risk factors, and these effects are likely much, much greater than anything gleaned from government restrictions.”

With the present data in hand, we cannot answer this question. However, in our opinion, the lack of these data does not invalidate our conclusions. Let us provide arguments for our stance:

 1) Of course, it is possible that some of these variables might be related to UO, but we are not aware of any empirical evidence that would point to them as critically important in all circumstances. The largest study which investigated this particular topic was conducted by Waters and colleagues (Waters et al., 2011). They found that health status was related to unrealistic pessimism (and not optimism). However, it is worth noting that the authors used a different method of measuring unrealistic optimism - they used comparisons to objective risk estimations, whereas we used subjective self/others comparison. This difference in measurement signals different definitions of UO/UP - Waters and collegues investigated being ‘unrealistic’ in an objective sense, while we ascribe purely subjective meanings to the labels “UO” or “UP”’.

More importantly, Waters’ research concerned a different and narrow context (breast cancer affects a small percentage of women - in our study the COVID-19 pandemic affects large portions of society) It is also worth mentioning that while in the aforementioned study unrealistic optimism was moderated by many demographic variables, unrealistic optimism was still present when these variables were controlled for.

 2) Because of our mistake in describing our key analysis as a ‘within-between’ subject ANOVA (for details see point 4 above), there might be a misunderstanding as to the main analytical basis of our conclusions. We want to clarify that we used no between-subject comparisons. We investigated the same 120 people throughout the whole study. For that reason, their individual baseline level of UO/UP has been accounted for. When we claim that there were differences in UO between waves or between periods of high and low restrictions, we mean that these differences were big enough to add a significant amount of the explained variance to the fraction of variance already explained by individual differences. 

 3) We agree that by not investigating more individual differences, we have missed the chance to discover potential moderators of our main effect, but in the absence of these moderating effects, we were still able to find the main effect.

 4) Our study was designed as a rapid response to the emerging, extremely unfamiliar circumstances, which called for a more focused and manageable design. Asking participants for medical information might require the approval of the bioethics committee and a different - stricter - approach to data collection. What is more, in order to avoid the pitfalls of self-description, we would need access to actual medical records.

 5) In reference to the previous point, such an addition would substantially shift our efforts and point of focus, which would result in a separate line of research - the relation between changes in health status and UO. While this is a very interesting and important issue, this was not our focus. 

Summing up - thanks to your suggestion we have made the following changes in the paper:

 • We have added a passage in the “Limitations and Directions for Future Research”, which describes the value of investigating individual characteristics (and changes in them) as possible mediators of the UO (see p. 23),

 • We have added a passage in the “Limitations and Directions for Future Research”, which underlines that the homogeneity of our research sample might influence the baseline of UO and diminish the generalizability of our conclusions.

We hope that the provided changes along with the provided rationale address your important comment. 

“It may also be that those with UO thought they were at lower risk during the government restrictions simply because they were not allowed to interact with others as much.”

This is an interesting possibility, and we also considered this interpretation.

Indeed, in our mixed analysis (see page 17-18), we found that the risk estimates for "Self" fell during times of greater restrictions. Estimates for "Others" also fell, but to a lesser extent. However, one critical issue remains unresolved: The ban on direct, interpersonal contact applied to all citizens, and our participants were fully aware of this. Despite this fact, they estimated risks as if these restrictions reduced their own risks more than those of others. This phenomenon calls for a more nuanced explanation - one that takes into account not only changes in the objective situation of the subjects but also biases in the interpretation of this change for themselves and others. Motivational or cognitive explanations, which we present on p. 13 and 19-21, seem more appropriate here.

“And I remain unconvinced that Google mobility trends are a proxy for amount of social contact.” 

We understand your reservations about this method - we are aware that this measure is not without its drawbacks. It is also a novel tool and for this reason, has not yet been thoroughly tested. However, we still believe that for our research purposes this measure can be sufficient and better than the self-reporting alternative. More importantly, in response to your great comment, we have rewritten the paragraph on this method to better explain our rationale behind this choice. (See p. 16). 

Additionally, we added some potential limitations of this method in the “Limitations and future directions” section (see p. 23)

“Overall this is a great idea and I appreciate the effort; the execution does not substantiate the conclusions at this time without more information on participant health status.”

Thank you again for your insightful comments and effort in evaluating our work. We hope that the changes and explanations we offer have elevated the overall value of our paper. We assure you that we are keen on improving the text further if you would find it advisable.

---

## [Decision Letter · Decision Letter 1]

9 Nov 2022

Temporal aspects of unrealistic optimism and robustness of this bias: a longitudinal study in the context of the COVID-19 pandemic

PONE-D-22-05224R1

Dear Dr. Izydorczak,

We’re pleased to inform you that your manuscript has been judged scientifically suitable for publication and will be formally accepted for publication once it meets all outstanding technical requirements.

Kind regards,

Maurizio Fiaschetti

Academic Editor

PLOS ONE

Reviewers' comments:

Reviewer's Responses to Questions

**Comments to the Author**

1. If the authors have adequately addressed your comments raised in a previous round of review and you feel that this manuscript is now acceptable for publication, you may indicate that here to bypass the “Comments to the Author” section, enter your conflict of interest statement in the “Confidential to Editor” section, and submit your "Accept" recommendation.

Reviewer #2: All comments have been addressed

Reviewer #3: All comments have been addressed

2. Is the manuscript technically sound, and do the data support the conclusions?

Reviewer #2: Yes

Reviewer #3: Yes

3. Has the statistical analysis been performed appropriately and rigorously? 

Reviewer #2: Yes

Reviewer #3: Yes

4. Have the authors made all data underlying the findings in their manuscript fully available?

Reviewer #2: Yes

Reviewer #3: Yes

5. Is the manuscript presented in an intelligible fashion and written in standard English?

Reviewer #2: Yes

Reviewer #3: Yes

6. Review Comments to the Author

Reviewer #2: Thank you for letting me read this paper and learn from the study described.

In this article the Authors present a very intriguing research concerning unrealistic optimism. For 12 months 120 participants were consequetivelly asked to estimate their own and their peers’ risk of COVID-19 infection. In this longitudinal study, the Authors were able to observe the robustness of the UO bias and to investigate the relations between the level of threat (objective information about the numer of cases and deaths as well as information about political decisions) and the magnitude of unrealistic optimism.

Overall, I found the paper very interesting to read. The manuscript is clearly written and the pattern of results obtained is intriguing. The theory and results obtained are relevant and contribute to the current body of knowledge, and this is why I think that after some (very minor) alternations are made to it, the article should be accepted.

(1) At the very begining of the article, the Authors introduce the term ”unrealistic optimism”. As authors most probably know, in psychology this term is used in two meanings:

- unrealistic absolute optimism

- unrealistic comparative optimism

(see: Shepperd, Klein, Waters & Weinstein, 2013 (Taking stock af unrealistic optimism – Perspectives on Psychological Science)

The authors decided to investigate comparative unrealistic optimism in their research (and without any doubts they were right, only such a measure is justified here). However, it is worth writing in the paper some words about the absolute optimism phenomenon (just 2 – 3 sentences). Thanks to this, the reader will have no doubts how the authors treated the unrealistic optimism and what they actually measured.

(2) I am very impressed by the authors longitudinal research on the dynamics of the unrealistic optimism. As far as I know, nobody in the world has investigated optimistic bias so many times in such a long period (one year!). Authors have shown changes in the level of unrealistic optimism under the influence of various factors.

The question came to my mind to what extend the level of unrealistic optimism is stable in normal (typical) situations. For ecample: does the level of unrealistic otpimism in relations to cancer or being the victim of car accident change over time? Of course, I am aware that the Authors do not know the answer to this question (I do not know it either), but perhaps it is worth asking such a rhetorical question in the section ”Limitation and directions of further research”.

Reviewer #3: I think the author did their best to address all comments and the manuscript is getting better and it could be published

7. PLOS authors have the option to publish the peer review history of their article (what does this mean?). If published, this will include your full peer review and any attached files.

Reviewer #2: No

Reviewer #3: No

---

## [Editor Report · Acceptance letter]

17 Nov 2022

PONE-D-22-05224R1 

Temporal aspects of unrealistic optimism and robustness of this bias:
a longitudinal study in the context of the COVID-19 pandemic 

Dear Dr. Izydorczak:

I'm pleased to inform you that your manuscript has been deemed suitable for publication in PLOS ONE. Congratulations! Your manuscript is now with our production department. 

Kind regards, 

on behalf of

Dr. Maurizio Fiaschetti 

Academic Editor

PLOS ONE